# Single-Case Study of Appetite Control in Prader-Willi Syndrome, Over 12-Years by the Indian Extract *Caralluma fimbriata*

**DOI:** 10.3390/genes10060447

**Published:** 2019-06-12

**Authors:** Joanne Griggs

**Affiliations:** Institute of Health and Sports, College of Health and Biomedicine, Victoria University, P.O. Box 14428, Melbourne, VIC 8001, Australia; joanne.griggs@vu.edu.au; Tel.: +61-399-192-203

**Keywords:** Prader-Willi syndrome, appetite treatment, *Caralluma fimbriata* extract, single-case

## Abstract

This paper reports on the successful management of hyperphagia (exaggerated hunger) in a 14yr-old female with Prader–Willi syndrome (PWS). This child was diagnosed with PWS, (maternal uniparental disomy) at 18 months due to developmental delay, hypertonia, weight gain and extreme eating behaviour. Treatment of a supplement for appetite suppression commenced at 2 years of age. This single-case records ingestion of an Indian cactus succulent *Caralluma fimbriata* extract (CFE) over 12 years, resulting in anecdotal satiety, free access to food and management of weight within normal range. CFE was administered in a drink daily and dose was slowly escalated by observation for appetite suppression. Rigorous testing determined blood count, vitamins, key minerals, HbA1c, IGF-1 and function of the liver and thyroid all within normal range. The report suggests a strategy for early intervention against hyperphagia and obesity in PWS. This case was the instigator of the successful Australian PWS/CFE pilot and though anecdotal, the adolescent continues to ingest CFE followed by paediatricians at the Royal Children’s Hospital Melbourne, Victoria, Australia. Future clinical trials are worth considering, to determine an appropriate dose for individuals with PWS.

## 1. Introduction

Prader-Willi syndrome (PWS) establishes a disrupted appetite with complex physical, behavioural and intellectual issues at a prevalence of 1:15,000–1:30,000 [1,2]. The predominant difficulty in PWS is an exaggerated appetite and a complex physiology which causes obesity. The hypothalamic dysfunction in PWS may be similar to early onset obesity or hypothalamic obesity [3,4] however, in PWS the disturbed neuroendocrine physiology is due to simultaneous non-functioning genes of the paternal chromosome in the critical region 15q11.2–q13. In PWS the characteristic appetite phenotype is established over three main phases [5] through: (1) ‘failure to thrive’ (newborn); (2a) increased weight gain with minimal adjustment in caloric intake; (2b) increased interest in food and caloric intake with a switch to hyperphagia at the median of 54 months and (3) the propensity to excessive eating (mean age 8 years). Typically, hyperphagia includes an impaired satiety [6,7]. Similar to obesity reported due to mutations in the PCSK1 gene [8], PWS establishes hypogonadotropic, hypogonadism and growth deficits. However, in PWS the obsessional eating behaviours determines the PWS phenotype to be life-threatening. Hypothalamic disruptions, the extent of which are unknown, interact with physiological deficits including growth hormone deficiency. Hypotonicity also exacerbates obesity propensity by making exercise and energy balance difficult. As in non-syndromic polygenic obesity, low caloric intake is important; however, in PWS this is constant and life-long. A range of interventions are necessary to correlate with each phase; however, to date there is no prescribed intervention or treatment for the switch to hyperphagia. Research into appetite supplements is limited [9,10] and management guidelines suggest interventions of diet, exercise and supervised access to food. This is difficult due to individualized anxiety and obsessional behaviours. Typically the most utilized intervention is life-long familial limit setting and external food surveillance, which for many includes locked environments [11,12].

*Caralluma fimbriata* extract (CFE)—the intervention—is well known in Ayurvedic medicine. This shrub grows wild throughout India, Pakistan and Afghanistan [13] and has been ingested for centuries amongst populations as a natural appetite suppressant or as a vegetable substitute in times of famine [13,14]. It is reported that genetic deletion of small nucleolar RNA (SnoRNA) including the snord116 and snord115 contribute to the hyperphagia in PWS [15,16,17]. One of the most important recognized disruptions is within the hypothalamic pathways, which are disturbed due to disrupted SnoRNA115/HBII-52 transcription known to interact with transcription of the serotonin (5Hydroxytryptamine; 5-HT) 5-HT2c receptor. This receptor is involved in anorexigenic signalling within hypothalamic appetite pathways of the central nervous system (CNS). Previously we reported that an extract of *Caralluma fimbriata* (CFE) used as the intervention in this case study was involved in enhancing 5-HT2c receptor activity in a PWS animal model [18]. This activity conforms to a model of increased satiety.

This single-case reports on continued management of PWS hyperphagic behaviours for a 14 year-old female (M) who has exhibited appetite at the highest eating code level: 5 [5]. Unusually, chronic administration of the natural supplement, CFE—a powdered cactus succulent extract—has been successful in maintaining reduced interest in food and a normal weight, over a period of 12 years in individuals.

## 2. Case Study

This single female case-study was one of twins (2004). M’s gestation was quiet compared to her brother and the birth was at 37 weeks by a double breach caesarean to a polyhydramnios mother of 42years-old. Both twins recorded a high Apgar score of 8/10, though M was administered oxygen straight after delivery. At 2 h M had multiple tonic-clonic seizures, was hypothermic, hypoglycaemic, had divergent gaze, hyperflexia and clonus. She was transferred to intensive care. A magnetic resonance imaging (MRI) determined the newborn had extra axial space in the brain. M was administered phenobarbital for seizures and was nasogastric tube fed due to “failure to thrive”. The other twin was jaundiced, placed in special care and was discharged at 5 days. M was transferred to special care at 2 weeks and was discharged at 5 weeks still nasogastric tube fed. Even though the core features of PWS were apparent, including small size, hypotonicity (low muscle tone) and an inability to suck or swallow [19,20], chromosome analysis was only conducted to rule out common disorders. An electroencephalogram (EEG) scan showed abnormal spike wave activity and awkward postures were noted. The diagnosis was recorded at discharge as encephalopathy and suspected Cerebral Palsy, until further investigations. After discharge the parents stopped both the seizure medication and tube feeding at 8 weeks and M was fed by formula in an adapted bottle. No further tonic-clonic seizures were seen however, absence seizures continued till 10 years.

The parents devised an intervention program from 4 months [21] and M’s feeding became normalized at 9 months, weight gain was seen at 12 months, however unusually—for encephalopathy—weight was recorded on the 90-percentile whilst height remained on the 3rd-percentile on the general public early weight/height for age guidelines. Hunger was noticed and celebrated; due to past difficulties at 14 months, however, M continued to eat past the amount of her twin brother. Observed behaviours included exaggerated crying, tantrums and reddening of the face. The facial reddening was perhaps due to unexplained seizures. As the parents were unaware of PWS, they utilized the exaggerated hunger to help M ambulate. For example, food was placed on a coffee-table and shifted around to help M weight-bare and walk. M would follow food with the table’s assistance. When left to her own devices with food nearby, M would continue to eat and would not stop of her own accord (video recordings on request).

Diagnosis of PWS was made at 18 months due to M’s weight gain and eating behaviour. The genetic female molecular karyotype showed one long continuous stretch of homozygosity (17.5 Mb) along chromosome 15, suggestive of maternal uniparental disomy (UPD). Other typical identifying features included small hands and feet, dysmorphic facial features, light skin pigmentation and severe developmental delay. Non appetite related issues experienced by M were vision impairment, deteriorating teeth, scoliosis, early puberty, repetitive questioning, skin picking until 12 years and anxiety.

## 3. Treatment Methodology

At 29 months of age CFE was sourced by the parents. Careful dosage of the product Slimaluma was considered by direct communication with the manufacturers. CFE was given for breakfast daily, in tropical juice (to cover the organic taste) at half a capsule (250 mg). The treatment dose was raised by 250 mg increments as necessitated by signals of reduced satiation. The commercially recommended adult dose is 1000 mg/d to be taken at 2 × 500mg doses. In M’s case the full dose was taken at once and after 12 years (Table 1) is 2000 mg/d. Further increases may no longer be available.

## 4. Results

During the earliest years of the intervention the child volunteered that “she felt full”, and “was not hungry”, instead wanting to draw (video if requested). This was a significant adjustment in comparison to recorded previous appetite behaviours, which were utilized as an instigator for movement. From 2–4 years, satiety eventually reduced in increments noticed by an increased asking for food. Each time this was observed the dose was increased by 250 mg/d. Within a day of increased administration, decreased hunger was noted. The dose therefore followed a simple pattern: expressed hunger, adjustment of dose and observed satiety. This process continued over 12 years with the most recent 250 mg/d increase raising the dose to 2000 mg/d in 2018. Each time with the amount settled upon there has been sustained successful appetite behaviour and freedom around food. All foods have been allowed within the diet however, M has been educated on the nutritional and caloric values of foods. It was noticed that M naturally initiates drinking a daily allowance of water (1 L/d).

Other medications administered were fluoxetine (LOVAN 20 mg, 9–14 years) for anxiety and Growth Hormone Therapy. Though there is clear evidence to support Growth Hormone (GH) use in PWS, it had been deemed that M was allergic to GH as after administration of a low dose of GH at two timelines, M showed signs of an allergic reaction including fluid retention and dark, puffy eyes. There is no known link to the intervention.

In 2010, M ceased taking the CFE extract for 6 days. Obvious changes to behaviour were recorded, including, crying and tantrums due to food. These remonstrations were far stronger than that observed by the parents when deciding to increase the dose. After 6 days the parents resumed administering CFE and continued with a routine of a broad diet with restricted daily caloric intake to 60%. The indicators of distress or hunger ceased within days and satiety resumed with M forgetting to eat after three days of CFE intervention (videos of before and after available).

Interestingly, certain of M’s behaviours are not attenuated by CFE. These behaviours may be due to anxiety and not hyperphagia, as they occur when M reports no hunger. These behaviours are: Skin picking; asking and telling; feeding babies and animals and wanting to know the routine of the day. These behaviours are moved-on easily during CFE administration. 

Over the 12 years of administration there have been no adverse effects. The parents suggest administration of savory salted foods increases focus after ingestion. Thirst for water may be an unusual effect of CFE. Over the 12 years gastric emptying had not been an issue. In 2011 after 4 years and 6 months administration of CFE, M was administered blood tests to determine any undue toxicity or adverse effect due to CFE administration. The measures noted in Table 2, including kidney, thyroid and liver function, blood count, IGF and HbA1C—measure of hyperglycaemia, were all within normal range. Endocrine tests for central adrenal insufficiency (CAI) also recorded levels within normal range. After 12 years, comprehensive blood tests determined M’s blood serum to be all within normal range (Table 2) apart from the lowering of triglycerides.

Weight reduced at the time of administration of CFE and had been stabilized to conform within a normal range. M’s diet remained at a 60% ratio (of someone the same weight), though food with lower sugar was preferable. Lipids seemed necessary within the diet to maintain homeostasis and salt maintained alertness. Exercise was preferred, though due to GH allergy, M’s muscle tone did not increase. Even so, M was active and energetic.

## 5. Discussion

This study establishes a case for management of hyperphagia by daily CFE administration in PWS. Over a 12-year-period hyperphagic behaviours have been contained without adverse effects in one child/adolescent with PWS. The prediction of obesity [22] has also been halted by this family’s management of dietary routine, exercise, supervision and CFE supplementation. The reduction of food has been accepted with continual adjustment in dose of CFE. Further, the adolescent’s weight has continued to decline over time. An important discussion point within this case-study is that continued maintenance of weight reduction could need a dose escalation past 2000 mg/d. The parents have chosen to longer increase the dose. Dose escalation has been gradual and though CFE is well tolerated, it is inevitable that the dose may continue to escalate until BMI and adolescent growth stop progressing. At this time, it has been confirmed that M’s growth is close to its ceiling, therefore the set dose may not be a problem. Though CFE has been deemed safe by multiple safety assessments [23] dose may be a limitation to this intervention’s capacity. Dose studies are necessary for a dose-weight ratio to be established. In the past Victoria University has piloted a successful study of CFE treatment in children and adolescents with PWS (*n* = 16), over a four-week period [10] —instigated by this anecdotal evidence. Future study will need to include adults with established hyperphagia; obesity and measures of glycaemic control would be preferable. Thirst may also be an important measure as M’s instinct to drink is unusual for individuals with PWS.

Another point to consider is that the management of hyperphagic behaviours in PWS are quite taxing on families. In M’s case her behaviours were similarly taxing during the 6 days without CFE in 2010, where vehement hunger was expressed. It is not clear if the hyperphagic distress experienced by M over the short experimental cessation of CFE was her natural hyperphagia or a stronger hyperphagia due to the years of satiety. Unfortunately, this connotes, anecdotally that there is no residual effect of CFE over time.

## 6. Conclusions

In conclusion, this single-case determined that an extract of the Indian cactus succulent *Caralluma fimbriata* eased hyperphagic over a 12-year-period. When the treatment was stopped for 6 days, excessive hunger returned. Therefore, anecdotally CFE administration appears to create abstinence from food within free access, leading to a routine natural cycle of appetite homeostasis, including both hunger and satiety. Unfortunately, CFE does not create a residual effect and dose-weight ratios must be defined to reduce confusion around dose adjustments to attenuate appetite behaviours. Even though this family aimed for independence without supervision for their daughter, dose escalation and phenotypical difficulties may have hindered this goal. This single-case study suggests that CFE increases satiety and maintains weight overtime, within parameters of caloric restriction without adverse effects or compromising blood serum measures.

## Figures and Tables

**Table 1 genes-10-00447-t001:** Age and anthropometric measurements in single-case with Prader-Willi syndrome (PWS) ingesting *Caralluma fimbriata* extract (CFE) . Anthropometric measurements behavioural indications of hyperphagia and additional intervention/exercise, in a single case of a female with PWS ingesting CFE over 12 years. The case begins as a newborn and ceases at 14 years and 4 months of age. Weight in Kg—kilograms, height in cm—centimetres, nil—represents no intervention, dose in mg—milligrams, per d—day.

Age Yr/Month	Weight Kg	Height Cm	Dose Mg/d	Food/Hyperphagia	Indication	Other Intervention
Birth	2.35	Unsure	nil	Failure to thrive	Unable to suck	Tube fed, phentobarbatal
2.3 months	3.7	55	nil	Feeding slowly	Take out nasogastric tube	Feeding extremely slowly; adapted bottle
8 months	6.5	65	nil	Normalised appetite	Bottle fed	Feeding slowly with adapted bottle
12 months	8.2	68	nil	Normalised appetite	Bottle fed and soft food	Adapted bottle exercise program
18 months	12	72	nil	Increased hunger	Obese and ambulating after food	Diet 100% home exercise program
20 months	12.9	78	nil	Eating constantly	Obese, always hungry ambulating after food	DIAGNOSIS
2 years	12.5	82.5	nil	Hyperphagia	Obesity, tantrums around food and always hungry	Diet 60% home exercise program
2 years, 4 months	12	85	250	Satiated	Saying no to dinner and leaving food on plate	Diet 60% home exercise program
4 years, 11months	19	104	500	Access to food with supervision	Saying no to dinner and minimal asking for food.	Diet 60% home exercise program
6 years, 6 months	23	112	NIL for 6 days	Ceased CFE hyperphagia returned after 1.5 days	Tantrums around food after 1.5 days. Licking empty plate	Diet 60% ballet once weekly
6 years, 6 months	23	112	500	Hunger persisted 1 day	Confirmation of satiety and interested in other activities	Diet 60% ballet once weekly
7 years, 1 month	23.5	114	750	Non-restricted with supervision	Both hunger and satiety	Diet 60% ballet once weekly
7 years, 9 months	25	119	1000	Non-restricted with supervision	Saying no to dinner and minimal hunger.	Diet 60% ballet once weekly
8 years, 8 months	29	124	1000	Non-restricted with supervision	Interested in food	Diet 60%, routine no exercise
9 years, 7 months	30	126	1250	Non-restricted with supervision	Forgetting to eat food provided	Diet 60%, routine no exercise
10 years, 1 months	34	128	1250	Non-restricted with supervision	Minimal communication of hunger.	Diet 60%, routine no exercise
11 years, 8 months	37	129	1250	Non-restricted with supervision	Minimal communication of hunger and throws out school treats.	Diet 60%, routine ballet once weekly
12 years, 5 months	38	134	1500	Non-restricted with supervision	Minimal communication of hunger. Makes own lunch.	Diet 60%, routine swimming weekly
13 years, 1 month	39	136	1750	Non-restricted; attempting no supervision	Communication of both hunger, “I’m hungry”, and satiation “I’m full”.	Diet 60%, routine swimming weekly
14 years, 4 months	43	141	2000	Non-restricted environment.no supervision	Communication hunger and takes self out of difficult situations.	Diet 60%, self-managed swimming weekly

**Table 2 genes-10-00447-t002:** Blood serum measures after 12 years of treatment *Caralluma fimbriata* extract (CFE) intervention. Blood serum measures after 12 years of treatment CFE ingestion, in a single-case of a 14-year-old female with Prader-Willi syndrome (PWS). IFCC - International Federation of Clinical Chemists; NGSP - National Glycohemoglobin Standardization Program; Hct – haematocrit; RCC – red cell count; MCV – mean corpuscular volume; MCH – haemoglobin devided by the number or red cells; RDW – red blood cell width and distribution; EDTA as an anticoagulant, INR—international normalized ratio and APTT—activated partial thromboplastin.

Specimen Type/Time 13:52	Serum Value	Reference Range
25-OH Vitamin D	65	50–160 nmol/L
Vitamin A	1.2	0.9–2.5 µmol/L
Vitamin E	21	13–24 µmol/L
Calcium	2.27	2.10–2.60 nmol/L
Magnesium	0.70	0.70–1.20 nmol/L
Phosphate	1.45	1.10–1.80 nmol/L
Cholesterol	3.7	3.1–5.4 nmol/L
Triglyceride	0.8 (L)	0.9–2.0 mmol/L
Vitamin E/Lipid ratio	4.8	0.9–7.1 umol/mmol
Active Vitamin B12(Holotranscobalamin)	49.7	19–128 pmol/L
Ferritin	36	9–136 µg/L
**Specimen type**	**Whole Blood EDTA**	
HbA1c (IFCC)	31	26–39 mmol/mol
HbA1c (NGSP)	5.0	4.5–5.7%
Red Cell Folate	2925	1800–3700 µmol/L
Haemoglobin	122	120–160 g/L
Hct	0.36	0.36–0.46
RCC	4.00	4.0–5.2 × 10^−12^/L
MCV	90	78–97 fL
MCH	30.4	25–33 pg
RDW	12.5	11.0–14.0%
Platelets	197	150–400 × 10^−9^/L
White Cell Count	7.3	4.5–13.5 × 10^−9^/L
Neutrophils	4.22	1.8–8.0 × 10^−9^/L
Lymphocytes	2.59	1.2–5.2 × 10^−9^/L
Monocytes	0.42	0.1–1.0 × 10^−9^/L
Eosinophils	0.07	0.0–0.5 × 10^−9^/L
Basophils	0.02	0.0–0.1 × 10^−9^/L
**Specimen**	**Serum plasma**	
Zinc	9.5	9.2–15.4 µmol/L
Selenium	1.0	0.6–1.9 µmol/L
Thyroid stimulating hormone	2.96	0.50–4.50 mlU/L
**Specimen**	**Blood plasma**	
INR	1.1	0.8–1.2 ratio
APTT	34	27–44 s
Fibrinogen	2.4	1.5–4.3 g/L
**Liver Function test**	**Serum**	
Total Bilirubin	2	0–15 unmol/L
Unconjugated Bilirubin	2	0–10 unmol/L
Bilirubin	0	0–5 unmol/L
Neutrophils	26	10–30 IU/L
Lymphocytes	117	100–350 IU/L
Monocytes	<10	0–40 IU/L
Eosinophils	72	57–80 g/L
Basophils	44	33–47 g/L
**Urea, Creatinine & Electrolytes**		
Sodium	140	135–145 mmol/L
Potassium	4.2	3.5–5.0 mmol/L
Chloride	102	98–110 mmol/L
Urea	5.4	2.1–6.5 mmol/L
Creatinine	34	30–80 µmol/L

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
