# Peer review of "Single-Case Study of Appetite Control in Prader-Willi Syndrome, Over 12-Years by the Indian Extract Caralluma fimbriata"

_genes, 2019, doi:10.3390/genes10060447_

Round 1

Reviewer 1 Report

Interesting single-case study that proposes the use of CF extract to prevent hyperphagia in PWS. I suggest several minor points to improve the manuscript:

1) I would include a more detailed introduction about the known genetic causes of obesity in PWS beyond deficiencies in the 5-H52C receptor. For instance, it has been recently reported a possible impact of PCSK1 defects on hypephagia an obesity in PWS patients (PMID:29880780)

2) Genetic alterations (molecular karyotype) used for diagnosis could be presented as supplementary material

3) Table 1 needs somme amendments:

- Age is not sufficiently described. Please add Yr or Mth after each number of the column for clarity.

- Please define NIL in the Table legend

4) Please add HbA1c levels (and any clinical parameters related to glucose metabolism if available) to Table 2.

Author Response

Thank you so much for your assessment.Details within the introduction now include a larger range of causes for obesity included the paper suggested.

The molecular karyotype has been cut for easier reading.

Amendments have been introduced within table 1 for clarity. HbA1c has been added to table 2 as a glucose parameter. There are no others available. 

The paper has also been cut for easier reading. 

Regards, 

Dr. Joanne Griggs

Reviewer 2 Report

This is a single patient report of a on the successful management of hyperphagia in a 14yr-old female with Prader–Willi syndrome (PWS). She received an Indian cactus succulent Caralluma fimbriata extract over 12yrs. I found this of interest. The manuscript is well written. In general: a lot of information is given on a single patient. This should be made shorter. The author is the parent of the patient presented. Has the patient given a concent to publication?

Line 16 in abstract: Rigorous testing determined blood serum levels within normal range. Serum levels of what? – It’s explained in the text but should be made clearer in the abstract.

Author Response

Thank you for your assessment. 

I have cut the paper down, however the other reviewer did ask for more detail within the introduction. This has been included. I have also added more information within the abstract regarding blood serum measurements.

The consent form has been submitted. 

Regards, 

Dr Joanne Griggs
